# Functions of Small Non-Coding RNAs in *Salmonella*–Host Interactions

**DOI:** 10.3390/biology11091283

**Published:** 2022-08-29

**Authors:** Xia Meng, Mengping He, Pengpeng Xia, Jinqiu Wang, Heng Wang, Guoqiang Zhu

**Affiliations:** 1Jiangsu Co-Innovation Center for the Prevention and Control of Important Animal Infectious Diseases and Zoonoses, College of Veterinary Medicine, Yangzhou University, Yangzhou 225009, China; 2Joint International Research Laboratory of Prevention and Control of Important Animal infectious Diseases and Zoonotic Diseases of China, Yangzhou 225009, China; 3Joint International Research Laboratory of Agriculture and Agri-Product Safety, The Ministry of Education of China, Yangzhou 225009, China; 4Department of Animal Husbandry and Veterinary Medicine, Beijing Agricultural Vocational College, Beijing 102442, China

**Keywords:** sRNAs, *Salmonella*, host, interactions

## Abstract

**Simple Summary:**

In the process of infecting the host, *Salmonella* senses and adapts to the environment within the host, breaks through the host’s defense system, and survives and multiplies in the host cell. As a class of universal regulators encoded in intergenic space, an increasing number of small non-coding RNAs (sRNAs) have been found to be involved in a series of processes during *Salmonella* infection, and they play an important role in interactions with the host cell. In this review, we discuss how sRNAs help *Salmonella* resist acidic environmental stress by regulating acid resistance genes and modulate adhesion and invasion to non-phagocytic cells by regulating virulent genes such as fimbrial subunits and outer membrane proteins. In addition, sRNAs help *Salmonella* adapt to oxidative stress within host cells and promote survival within macrophages. Although the function of a variety of sRNAs has been studied during host–*Salmonella* interactions, many of sRNAs’ functions remain to be discovered.

**Abstract:**

*Salmonella* species infect hosts by entering phagocytic and non-phagocytic cells, causing diverse disease symptoms, such as fever, gastroenteritis, and even death. Therefore, *Salmonella* has attracted much attention. Many factors are involved in pathogenesis, for example, the capsule, enterotoxins, *Salmonella* pathogenicity islands (SPIs), and corresponding regulators. These factors are all traditional proteins associated with virulence and regulation. Recently, small non-coding RNAs (sRNAs) have also been reported to function as critical regulators. *Salmonella* has become a model organism for studying sRNAs. sRNAs regulate gene expression by imperfect base-pairing with targets at the post-transcriptional level. sRNAs are involved in diverse biological processes, such as virulence, substance metabolism, and adaptation to stress environments. Although some studies have reported the crucial roles of sRNAs in regulating host–pathogen interactions, the function of sRNAs in host–*Salmonella* interactions has rarely been reviewed. Here, we review the functions of sRNAs during the infection of host cells by *Salmonella*, aiming to deepen our understanding of sRNA functions and the pathogenic mechanism of *Salmonella*.

## 1. Introduction

*Salmonella enterica* is one of the leading causes of foodborne gastroenteritis worldwide. The two most important serovars of *Salmonella* are *Salmonella enterica* serovar Typhimurium (*S.* Typhimurium) and *Salmonella enterica* serovar Enteritidis (*S.* Enteritidis), which cause non-typhoid salmonellosis infections [1]. As an intracellular zoonotic pathogen, *Salmonella* regularly infects hosts. It enters the stomach and intestinal lumen of the host after ingestion of contaminated food, causing gastroenteritis in both humans and animals as well as typhoid fever in mice. *Salmonella* must survive within the acidic environment of the stomach and penetrates the gut barrier via M cells in Peyer’s patches of the intestine [2]. *Salmonella* invades the cell membrane and forms *Salmonella*-containing vacuoles (SCVs) with the help of a Type III secretion system (T3SS) encoded by *Salmonella* pathogenicity islands (SPIs) [3]. After that, macrophages engulf the bacteria and kill them to resist infection by producing reactive nitrogen species (RNS) and reactive oxygen species (ROS) [4]. Interestingly, *Salmonella* employs sophisticated strategies to survive and replicate inside phagocytic and non-phagocytic cells, causing serious diseases in humans and animals.

The small non-coding RNAs (sRNAs), which are known to be involved in the regulation of gene expression, have a length of 50–500 nucleotides and have been found in various bacteria, for example, *Escherichia coli*, *Listeria monocytogenes*, and *S.* Typhimurium [5,6]. Based on their mode of base-pairing, they are classified into cis- and trans-encoded sRNAs. Cis-encoded sRNAs are transcribed from the same loci as the mRNAs on the opposite strand of DNA and bind to their cognate mRNA targets with perfect complementarity, resulting in either transcriptional termination or translational initiation. Trans-encoded sRNAs interact with multiple mRNA targets through imperfect complementation [6,7]. Gene expression is usually regulated by trans-acting sRNAs at the post-transcriptional level [8]. The functions of more than half of the trans-acting sRNAs require the chaperone protein Hfq, which plays an important role in regulation by stabilizing sRNAs and mediates their interaction with the trans-encoded target mRNAs of host cells, leading to repression of translation or acceleration of mRNA decay [9]. *S.* Typhimurium expresses hundreds of sRNAs, many of which are activated under special stress and virulence conditions, suggesting that sRNAs are an important component of regulatory networks controlling gene expression in bacteria during host infection [10].

sRNAs regulate many physiological processes in bacteria, including metabolism, iron homeostasis, quorum sensing, outer membrane protein biosynthesis, and the regulation of virulence genes [11,12]. In recent years, attention has been focused on the functions of sRNAs in bacteria–host interactions. To establish a successful infection, *Salmonella* must first resist the acidic environment and oxidative stress, adhere to and invade non-phagocytic cells, and finally evade host immunity to survive inside macrophages [13]. sRNAs play integral roles in bacterial stress responses, promote intracellular survival, and modulate host immune responses [9,14]. In this review, we summarize the roles of sRNAs in the interaction between *Salmonella* and host cells (see Table 1 for a summary of sRNAs), aiming to understand the roles of sRNAs upon host cell infection, provide an overview of the functional mechanisms of sRNAs, and provide ideas to improve host resistance to *Salmonella* infection.

## 2. sRNAs’ Functions in *Salmonella*–Host Interactions

### 2.1. sRNAs Regulate Resistance to the Acidic Environment

Acid stress is one of the most important stresses that *Salmonella* must overcome to colonize the host. *Salmonella* may encounter acidic environments such as the stomach, and acid exposure also occurs after invading the intestinal mucosa [13,36]. Interestingly, *Salmonella* has evolved a precise response known as the acid tolerance response (ATR), which promotes the survival of acid-adapted cells at low pH levels [36]. The alternative sigma factor RpoS, which is a global regulator of gene expression when bacteria suffer from starvation or other stress conditions, is required for *Salmonella* resistance to acid stress [37]. Recently, sRNAs have been identified as major regulators of acid stress response networks. The 87-nucleotide sRNA DsrA, which is present in both *Salmonella* and *E. coli*, is a regulator of acid resistance. DsrA expression can be induced in *S.* Typhimurium in a minimum essential medium, with maximum induction at pH 3.1. Deletion of *dsrA* reduced the effectiveness of the ATR, resulting in a lower survival rate in acidic environments [38]. Although how DsrA modulates the acid stress response is still unclear, some evidence has proven that DsrA could regulate the translation of RpoS and the expression of some acid resistance genes [15,39,40]. DsrA activates the translation of RpoS by base-pairing with the upstream leader portion of *rpoS* mRNA in *S*. Typhimurium [15]. Mutations in *dsrA* decreased the expression of RpoS in exponential and stationary phases of *E. coli* [39]. The mRNA levels of multiple acid resistance genes in the *hdeAB*, *gadAX*, and *gadBC* operons were increased with DsrA overproduction in *E. coli* [40]. The above reports are useful for further study of the mechanism of acid resistance regulation through DsrA. Besides the acid stress response, DsrA enhances the invasive phenotype of *S.* Typhimurium by repressing the translation of the histone-like nucleoid protein (H-NS) [15,38]. Moreover, deletion of *dsrA* reduced the motility, adhesion, and invasion efficacy of *S*. Typhimurium [38]. Another example is the sRNA RyeC, which is the antitoxin component of a type I toxin–antitoxin (TA) system that is encoded by a small ORF and is associated with extremely high toxicity [41]. Overexpression of the trans-encoded RyeC in *S.* Typhimurium during the ATR inhibits the expression of the target PtsI, which is a subunit of a major carbohydrate transporter, through direct interaction at the post-transcriptional level, resulting in a reduced ATR in *Salmonella* [16].

Survival in acidic environments is essential for *Salmonella* to cause infection. 6S RNA, which is a highly conserved sRNA in prokaryotes encoded by *ssrS*, promotes *S.* Typhimurium survival at pH 3.0. Deletion of 6S RNA in *S*. Typhimurium decreases the expression of the *citGXFED* and *nuo* operon, which is related to citrate metabolism and encodes the energy-conserving NADH dehydrogenase, respectively [17]. It has been found that the genes of the *nuo* operon were significantly upregulated under acid stress in *S*. Typhimurium [42], and the electron transport system involved in NADH dehydrogenase can resist acid stress by coupling proton efflux to energy generation [43]. 6S RNA may help *Salmonella* to resist acid stress, largely by upregulating the *nuo* operon. Additionally, 6S RNA also increases the ability of *S.* Typhimurium to invade epithelial cells, playing an important role in virulence [17].

### 2.2. sRNAs Regulate Adhesion to and Invasion of Non-Phagocytic Cells

*Salmonella* initiates infection of the host by attaching to the intestinal mucosal surface and subsequently adhering to and invading non-phagocytic cells. Adhesion to and invasion of host cells are crucial steps in *Salmonella* infection. Many adhesive structures such as fimbrial and non-fimbrial proteins were found in *Salmonella*, including type I fimbriae, autotransporter adhesins, and the outer membrane protein OmpD [1,44,45,46].

#### 2.2.1. sRNAs Regulate the Expression of a Fimbrial Subunit

FimA is the major fimbrial subunit of Type 1 Fimbriae in *Salmonella*, and it is important for adhesion to enterocytes and colonization of the intestine [18]. STnc640, which is an Hfq-binding sRNA identified in *S.* Typhimurium by deep sequencing [47], is also present in *S.* Enteritidis and upregulates the expression of *fimA*. Although STnc640 regulates the expression of *fimA*, it decreases the adhesion ability of *S.* Enteritidis to human colorectal adenocarcinoma epithelial cells (Caco-2) and attenuates the virulence of *S.* Enteritidis in chickens [19]. STnc150, another sRNA of *S.* Typhimurium, downregulates the protein expression of FimA by base-pairing with the 5’-end of *fimA* mRNA. Deletion of STnc150 enhanced the adhesion ability of *S.* Typhimurium to macrophages and reduced LD50 in mice [48].

#### 2.2.2. sRNAs Regulate the Expression of Outer Membrane Proteins

OmpD is the most abundant outer membrane protein in *Salmonella* and plays a significant role in adherence to human macrophages and intestinal epithelial cells [45]. MicC is an Hfq-associated sRNA that is expressed in *S.* Typhimurium and *E. coli*. It can regulate the expression of outer membrane proteins such as OmpD, OmpC, and OmpN [20,21,49]. MicC represses the expression of OmpD by base-pairing with the coding sequence of *ompD* mRNA and accelerating RNase E-dependent *ompD* mRNA decay [21]. Additionally, Hfq contributes to the annealing of MicC to the coding sequence of *ompD* mRNA and the induction of specific conformational changes [50]. Deletion of *micC* increased the expression of OmpD and subsequently enhanced the virulence of *S.* Enteritidis in mice and chickens [51]. InvR, a SPI-1-encoded sRNA, was directly activated by the major SPI-1 transcription factor HilD under invasion-inducing conditions. The stability of InvR was dependent on Hfq in vivo. InvR can also repress the synthesis of OmpD [22]. Deletion of *invR* enhanced the adhesion ability of *S.* Enteritidis to Caco-2 cells and increased the pathogenicity of *S.* Enteritidis to chickens [52].

#### 2.2.3. sRNAs in SPIs and Regulation of SPI Genes Related to Invasion

The virulence of *Salmonella* is largely due to SPIs. SPIs are chromosomal regions that contain many virulence-related gene clusters, such as fimbriae-related operons, T3SSs, and associated transcriptional regulatory factors [53]. Besides the protein-coding genes of SPIs, there are numerous non-coding sRNAs encoded within the SPIs of *Salmonella*. Padalon-Brauch et al. identified and characterized 19 unique sRNAs from the SPIs of *S.* Typhimurium. Several of these sRNAs responded to stress conditions and were expressed in the stationary phase [23]. For example, IsrC was detected under stress conditions such as acidic conditions (pH 4.9), oxidative stress, iron-limited conditions, osmotic shock, and heat shock. Isrk and IsrJ were expressed during the late stationary phase in cells grown under low-oxygen or low-magnesium conditions. IsrP was expressed under low-magnesium and extremely acidic conditions (pH 2.5). These sRNAs might contribute to invasion of and survival within macrophages [23].

Several SPI-encoded sRNAs affect the invasion of non-phagocytic cells by *Salmonella*. For instance, IsrJ, an sRNA positively regulated by HilA, facilitates bacterial invasion into the HeLa cells. HilA is a transcriptional activator that responds to environmental signals and regulates the transcription of SPI-1 genes. Notably, deletion of *hilA* leads to a loss of invasion ability of *Salmonella* into epithelial cells. In addition, IsrJ promotes the translocation efficiency of T3SS-1 effector protein SptP into eukaryotic cells [23]. IsrM, an SPI-encoded sRNA, is shown to promote invasion and intracellular replication by regulating the expression of virulence-related factors within SPI-1. The expression of IsrM is induced not only under conditions resembling the gastrointestinal tract in vitro but also during the early and late stages of *Salmonella* infection in vivo, with higher expression in the ileum than in the spleen. IsrM targets the 5’ untranslated regions (UTRs) of the mRNA encoding the SPI-1 effector SopA and the global SPI-1 regulator HilE [24]. HilE negatively regulates the expression of many SPI-1 genes. SopA is essential for bacterial invasion. Both are major virulence factors. Deletion of *isrM* leads to the dysregulation of SopA and HilE, as well as the genes regulated by HilE that are present in SPI-1. This subsequently decreases the ability of *Salmonella* to invade epithelial cells and to survive or proliferate in macrophages. In addition, mutation of *i**srM* causes a defect in *Salmonella* growth in the ileum and spleen of BALB/c mice and attenuates the pathogenesis of *Salmonella* in mice [24].

IsrE (also known as RyhB-2) is an SPI-encoded sRNA whose expression is induced under iron-limited and starvation conditions [23]. *Salmonella* carries two RyhB homologs, RyhB-1 and RyhB-2. Both are expressed under low-iron conditions. Besides regulating iron acquisition and homeostasis [54], RyhB-1 and RyhB-2 directly upregulate the expression of T3SS effectors SipA and SopE via incomplete base-pairing [25]. Two T3SSs have been identified in *Salmonella*, T3SS-1 and T3SS-2, which are encoded by SPI-1 and SPI-2, respectively. As major virulence determinants, the T3SS injects various effector proteins into the host cytoplasm. This is essential for the infection process in host cells and for the pathogenicity of Gram-negative bacteria [55,56]. SipA and SopE, which are T3SS effector proteins, facilitate *Salmonella* invasion of non-phagocytic cells. SipA and SopE can be delivered using outer membrane vesicles (OMVs) as vehicles into host cells. This process is independent of *Salmonella*–host cell contact [57]. Then, SipA elicits host cellular actin polymerization and activates caspase-3 in both intestinal epithelial cells and macrophages, which contributes to invasion of epithelial cells and survival at the early stages of *S.* Typhimurium infection [58]. SopE promotes entry into host cells by triggering actin-dependent ruffles and contributes to intracellular replication through transient localization to the early SCV [59]. RyhB-1 and RyhB-2 upregulate the expression of SipA and SopE and subsequently enhance the adhesion and invasion ability of *S.* Enteritidis in intestinal epithelial cells [25].

Interestingly, InvS, an 89-nucleotide sRNA, was first identified as STnc470 in *S.* Typhimurium [47]. InvS expression is induced under various stress conditions and positively regulated by two-component regulatory systems such as PhoP/Q, SsrA/B, and OmpR/EnvZ [60]. More importantly, InvS efficiently promotes *Salmonella* adherence to and invasion into non-phagocytic cells by coordinating the expression of the T3SS proteins PrgH and FimZ, which are regulators known to facilitate fimbrial protein expression. It is obvious that the sRNA InvS not only promotes invasion but also adhesion of *Salmonella*. In addition, InvS (i) regulates the secretion of the *Salmonella* T3SS effector proteins SipA, SipB, and SipC, and (ii) affects effector translocation during *Salmonella* infection [26].

PinT (also known as STnc440) is a PhoP-activated sRNA from a horizontally acquired *Salmonella*-specific locus [22]. The expression of PinT is dramatically upregulated during infection of HeLa cells and diverse macrophages, as well as in the SPI-2-inducing medium. PinT is a post-transcriptional regulator of three virulence-related systems: T3SS-1, T3SS-2, and the flagellar regulon. It represses the SPI-1 invasion-associated effectors SopE and SopE2 by base-pairing with the ribosome binding site and the transcription start site. Dual RNA-seq unveiled the functions of sRNAs in host–pathogen interactions. PinT also inhibits the expression of invasion genes globally by downregulating the translation of HilA (hyper-invasion locus A) and RtsA (a kind of AraC-like protein). It binds the mRNAs of *hilA* and *rtsA* directly by base-pairing, resulting in translational inhibition of HilA and degradation of *rtsA* transcripts [27]. HilA is a major transcriptional regulator of the T3SS, activating the transcription of T3SS structural components and effector proteins. HilD and RtsA control the T3SS by forming a complex feed-forward loop to activate the expression of HilA. The PhoP/Q two-component system, which is critical for intracellular survival, negatively regulates SPI-1 expression by directly repressing the transcription of *hilA* and indirectly repressing the transcription of both *hilD* and *rtsA* [61]. Therefore, PinT assists in phoPQ-mediated regulation of T3SS-1 at the post-transcriptional level [27]. For SPI-2 regulation, PinT directly represses the expression of GrxA (a component of the glutathione/glutaredoxin system) and synthesis of the transcription factor CRP (a cyclic AMP receptor protein), as well as translation of SsrB (a primary regulator of T3SS-2), which affects the activation of SPI-2. PinT is a timer that temporally shapes the transition from SPI-1 to SPI-2. In other words, PinT is involved in the switch from the invasion stage to intracellular survival [27,34]. Recently, a novel PinT target, SteC (secreted effector kinase), was identified using an approach called MS2 affinity purification and RNA-sequencing (MAPS). PinT blocks the synthesis of SteC by targeting the 5’-UTR of *steC* to inhibit translation initiation, resulting in suppression of SteC-mediated actin rearrangement in HeLa cells [35]. In addition, PinT represses motility by indirectly downregulating the expression of the flagellar genes *flhDC* and *fliZ* via CRP. Taken together, PinT is a critical post-transcriptional regulator for *Salmonella* virulence and adaptation to niches in the host.

#### 2.2.4. sRNAs in OMVs

OMVs have been proven to play crucial roles in various processes, such as stress responses and delivery of virulence factors during host–pathogen interactions. Many sRNAs have been found in OMVs. Some OMV-associated sRNAs involved in virulence, such as IsrA, IsrB-2, IsrD, and IsrM, were enriched in OMVs under different growth conditions, especially in SPI-1- or SPI-2-inducing conditions. Almost all SPI-encoded sRNAs have been found in OMVs. Some core genome-encoded sRNAs, such as MicF, CsrC, and GcvB, are also enriched in OMVs [62]. This indicates that these sRNAs are protected by OMVs in *S.* Typhimurium. This relationship between sRNAs and OMVs may promote host–pathogen interactions during infection.

### 2.3. sRNAs Regulate Resistance to Oxidative Stress within Cells

After entry into phagocytic cells, *Salmonella* faces oxidative stress in the cytoplasm. Phagocytic cells have two important antimicrobial systems, the NADPH phagocyte oxidase (phox) and inducible nitric oxide synthase (iNOS) pathways, which produce ROS and RNS, respectively. ROS play important roles in the early host response to infection, and RNS limit bacterial survival in host cells [63]. As a result, the ability of bacteria to survive the oxidative stress inside hosts is the key to induce pathogenicity. Interestingly, bacteria have involved mature mechanisms to promote survival and replication inside host cells. Recently, sRNAs in *Salmonella* have been demonstrated to have irreplaceable functions in resisting oxidative stress inside host cells. *S.* Enteritidis strain SE2472 can use mammalian atypical miRNA processing machinery to cleave sRNA into a ~22-nt miRNA-like RNA fragment, Sal-1, which facilitates the intracellular survival of invaded bacteria by targeting cellular iNOS, attenuating the iNOS-mediated antimicrobial ability of human colonic epithelial cells. sRNAs are important in the resistance to oxidative stress inside host cells [28].

OxyS is a stable and abundant sRNA that was first identified in *E. coli*. It has been proven that OxyS helps bacteria adapt to the mutagenic effects of hydrogen peroxide (H_2_O_2_) and protects *E. coli* against oxidative damage [64,65]. Moreover, OxyS is strongly activated when *S.* Typhimurium resides within J774 macrophages, suggesting that SPI-encoded sRNAs play significant roles in the network that regulates the stress response within the macrophage environment [23]. A recent report showed that OxyS could positively regulate the mRNA levels of the porin-encoding gene *ompX* under H_2_O_2_-induced stress in *S.* Typhimurium, and sRNAs CyaR and MicA positively regulate *ompA* mRNA levels under H_2_O_2_-induced stress [29]. To sum up, OxyS provides important assistance for bacteria to resist oxidative stress.

Some sRNAs also target virulence genes under oxidative stress to promote survival and replication in macrophages. RaoN is a sRNA encoded in the *cspH*-*envE* intergenic region on SPI-11. The expression of RaoN is increased under oxidative stress with nutrient-limiting conditions in vitro. It has been shown that the lactate dehydrogenase gene *ldhA*, whose inactivation and overexpression both render *Salmonella* more sensitive to oxidative stress, was upregulated in the *raoN* knockout mutant. Deletion of *raoN* impedes *Salmonella* survival and replication in macrophages [30,33].

RyhB and its homologs are key regulators of iron homeostasis in *E. coli* and *Salmonella* [31,66]. As an iron-regulatory sRNA, RyhB downregulates a large number of transcripts encoding iron-using proteins when facing iron deficiency and modulates the usage of intracellular iron. Beyond that, RyhB-1 and RyhB-2 expression is induced upon exposure to H_2_O_2_ [23]. RyhB deletion mutants (Δ*ryhB-1*, Δ*ryhB-2*, and Δr*yhB-1*/Δ*ryhB-2*) displayed increased levels of intracellular ROS and a growth defect when treated with H_2_O_2_ in iron-rich or iron-deficient conditions. OxyR upregulated the expression of *ryhB-1* and *ryhB-2* through direct interaction with their promoter region when *Salmonella* was treated with H_2_O_2_ [67]. The above illustrates that RyhB-1 and RyhB-2 are necessary for *Salmonella* to resist oxidative stress in the intracellular environment.

### 2.4. sRNAs Regulate Survival in Macrophages

Though T3SS-1 effectors are translocated into host cells to promote invasion of diverse cell types, the expression of T3SS-2 (encoded by SPI-2) is mainly triggered after entering cells and results in translocation of effectors to manipulate the intracellular niche [46,68]. *Salmonella* suffers extreme stresses in macrophages because macrophages produce RNS and ROS to kill it [4]. Therefore, survival and replication of *Salmonella* within macrophages is essential for its pathogenicity in hosts. Recently, many sRNAs have been found to promote survival and replication of *Salmonella* in macrophages.

When *Salmonella* is present in the intracellular environment of macrophages, the expression of many sRNAs is induced. Transcriptome analysis of intra-macrophage *S.* Typhimurium showed that 88% of 280 sRNAs were expressed [69]. Compared to the early stationary phase (ESP) conditions, with high expression of SPI-1 genes, 34 sRNAs (including RyhB-1, RyhB-2, OxyS, MicF, and RybB) were upregulated, and 119 sRNAs were downregulated [68]. The expression of many SPI-encoded sRNAs is induced when *Salmonella* multiplies within macrophages. For example, the transcription of IsrC and IsrN is increased 7-fold within the first hour post-infection of J774 macrophages and declines as infection progresses. IsrC overlaps with its flanking gene *msgA* at the 3’-end and affects the expression of *msgA* (encoding a macrophage survival-related protein) in cis. This indicates that IsrC is important for *Salmonella* to survive in macrophages [23]. OxyS, an sRNA that responds to oxidative stress, shows the same expression pattern as IsrC and IsrN. OxyS levels increase 35-fold within the first hour of infection and decrease thereafter. OxyS is strongly activated by H_2_O_2_ and increases the H_2_O_2_ resistance of *Salmonella* in macrophages [23]. The transcript levels of IsrH, IsrE (RyhB-2), and its homolog RyhB-1 increase within the first hour of infection and then increase dramatically at 8 h post-infection. The induction of expression of these sRNAs suggests that these sRNAs play important roles in the survival and replication of *Salmonella* inside macrophages [23].

Besides regulating the invasion of epithelial cells and the response to oxidative stress, RyhB-1 and RyhB-2 also contribute to intracellular survival in macrophages. In *S. Typhi*, RyhBs (RfrA and RfrB) are regulated by the ferric uptake regulator (Fur). Expression of both is induced within the human monocyte cell line THP-1 at 24 h post-infection. They are crucial for *Salmonella* replication inside macrophages [31]. In *S.* Typhimurium, RyhB-1 and RyhB-2 are the most highly induced sRNAs within macrophages compared to ESP conditions [69]. They contribute to survival and proliferation inside RAW264.7 murine macrophages [32]. Deletion of RyhB-1 and RyhB-2 leads to an increase in ATP levels and a decrease in the NAD+/NADH ratio, resulting in more active metabolism of *S.* Typhimurium in macrophages. RyhB-1 and RyhB-2 affect the expression of SPI-1-related genes encoding the transcriptional regulators HilA, HilC, HilD, InvF, and RtsA, the effector proteins SipA, SipB, SopA, and SopB, and the tricarboxylic acid cycle-related genes *fumA* and *sdhD*, which encode iron-containing enzymes involved in energy metabolism within macrophages. Moreover, RyhB-1 and RyhB-2 directly downregulate the expression of *rtsB* (encoding an invasion chaperone) and *sicA* (encoding a regulatory protein). This suggests that the two sRNAs possess the ability to integrate a global response to multiple stresses encountered inside macrophages [32].

IsrM, an SPI-encoded sRNA, influences bacterial invasion by independently targeting *hilE* mRNA. Moreover, the Δ*isrM Salmonella* mutant strain exhibits a defect in intracellular replication in vitro, upregulating the protein expression of SopA, confirming that it also enhances the survival and proliferation of *Salmonella* inside macrophages [24]. RaoN reduces the ROS levels in macrophages by reducing the gene expression of l*dhA*, which is conducive to the intracellular survival of *S.* Typhimurium. RaoN can also help to resist nitrosative-oxidative stress to enhance the virulence and survival of *S.* Typhimurium in vivo [29,33].

### 2.5. sRNAs Regulate the Expression of Inflammatory Cytokines in Host Cells

The innate immune system is essential to defend against bacterial pathogens [70]. SCVs induce the innate immune response through the interaction of pathogen-associated molecular patterns with Toll-like receptors in host cells, such as epithelial cells and macrophages. Upon this interaction, mitogen-activated protein kinases produce a signal, leading to secretion of a variety of cytokines, such as tumor necrosis factor and interleukin-8 (IL-8), which stimulate antibacterial responses and are beneficial for bacterial clearance [14,71,72]. However, *Salmonella* has evolved sophisticated methods to modify the host cells to meet their needs by regulating these inflammatory cytokines. This regulatory function of sRNAs of *Salmonella* has received increasing attention in recent years.

Besides regulating the host cell invasion, PinT has been shown to impact the host response using dual RNA-seq. PinT affects the host transcriptome during the whole course of the infection [34]. Deletion of *pinT* in *S.* Typhimurium results in activation of a key regulator of the JAK–STAT signaling pathway, suppressor of cytokine signaling 3 (SOCS3), and a reduction in the phosphorylation of Signal Transducer and Activator of Transcription 3 (STAT3) in HeLa cells [34]. Dual RNA-seq also revealed that the mRNA and protein expression of the pro-inflammatory chemokine IL-8 in HeLa cells was increased when infected with the *S.* Typhimurium Δ*pinT* mutant. Since SopE and SopE2, which are controlled by PinT, affect JAK–STAT signaling and IL-8 secretion, *Salmonella* may manipulate these host cell responses and facilitate intracellular replication by PinT-controlled virulence genes [34,35].

## 3. Conclusions and Perspectives

When *Salmonella* is ingested into hosts, it must first pass through the extremely acidic stomach, invade epithelial cells of the small intestine to form SCVs, survive inside macrophages and neutrophils, and then spread into the spleen, liver, and other systemic sites of the body to cause infection [2,73,74]. When facing acidic environments, sRNAs can quickly respond to environmental stresses via up- or down-regulating gene expression. When invading non-phagocytic cells, sRNAs help *Salmonella* to enter host cells, although the exact underlying molecular mechanisms remain to be explored. Moreover, sRNAs regulate related genes, contribute to resistance to nitrosative-oxidative environments, and enhance *Salmonella* survival and replication within macrophages. More importantly, sRNAs regulate host immune responses to facilitate infection. An overview of all the above sRNA-mediated processes is provided in Figure 1.

Decades of research have led to the discovery of numerous molecular strategies during different phases of the bacterial infectious process. These strategies, which are beneficial to the establishment of successful infection, are the results of selection during the process of pathogen–host co-evolution. *Salmonella* is a model bacterium to study host–pathogen interactions [75], whose strategies to adapt to the host and survive inside host cells are gradually revealed. We summarized the regulatory roles of sRNAs in different processes during *Salmonella*–host cell interactions. This review provides ideas for researching the roles of sRNAs and lays a foundation for studying the relationship between *Salmonella* and hosts, making sRNAs potential targets for pathogen manipulation [24]. It is important to know that most sRNAs perform different regulatory functions throughout the whole process of infection by regulating virulence genes. For example, the two homologs of RyhB, RyhB-1 and RyhB-2, can not only promote invasion of non-phagocytic cells but also regulate the survival capability of *Salmonella* inside phagocytic cells [32,76].

It is clear that much remains to be elucidated with respect to the functions of sRNAs during the process of bacterial infection, especially from the perspective of regulating host immune responses. Therefore, the interactions between sRNAs and host cells deserve more attention. Regulatory sRNAs are potential targets for pathogen manipulation, and the identification of more sRNAs will (i) improve our understanding of *Salmonella*–host interactions and (ii) provide new research directions.

## Figures and Tables

**Figure 1 biology-11-01283-f001:**
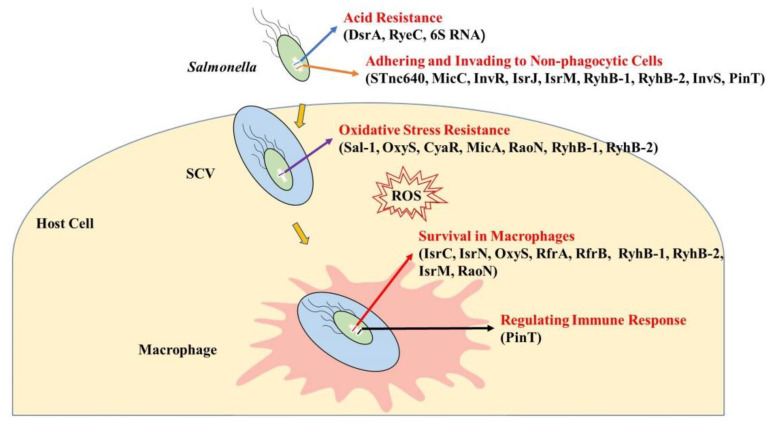
Regulatory sRNAs during the processes of *Salmonella* infecting host cells. When entering hosts, *Salmonella* encounters an acidic environment in the stomach. sRNAs DsrA, RyeC, and 6S RNA help *Salmonella* resist acid stress. In the process of adhering and invading to non-phagocytic cells, STnc640, MicC, InvR, IsrJ, IsrM, RyhB-1, RyhB-2, InvS, and PinT improve the adhesion and invasion ability of *Salmonella* by upregulating the expression of adhesins and invasion factors. When encountering intracellular oxidative stress, Sal-1, OxyS, CyaR, MicA, RaoN, RyhB-1, and RyhB-2 assist *Salmonella* to overcome this harsh condition. More importantly, sRNAs IsrC, IsrN, OxyS, RfrA, RfrB, RyhB-1, RyhB-2, IsrM, and RaoN promote the survival and replication of *Salmonella* in macrophages. PinT also helps *Salmonella* manipulate host cell immune responses by regulating the expression of inflammatory cytokines in hosts.

**Table 1 biology-11-01283-t001:** Summarized characteristics of sRNAs during *Salmonella*–host interactions.

Function in Infection	Serotype	sRNA	Description	Target Gene/Protein	Reference
Resisting AcidEnvironment	*S.* Typhimurium SB300	DsrA	Trans-coded	*rpoS*	[15]
	*S.* Typhimurium	RyeC	Trans-coded	*ptsi*	[16]
	*S.* Typhimurium	6S RNA	Trans-coded	*citGXFED*, *nuo* operon	[17]
Adhering and Invading to Non-Phagocytic Cells	*S.* Enteritidis 50336	STnc640	Trans-coded	*fimA*	[18,19]
	*S.* Typhimurium	MicC	Trans-coded	OmpD	[20,21]
	*S.* Enteritidis	InvR	Trans-coded	OmpD	[22]
	*S.* Typhimurium	IsrJ	Trans-coded	SptP	[23]
	*Salmonella*	IsrM	Trans-coded	HilE	[24]
	*S.* Enteritidis 50336	RyhB-1, RyhB-2	Trans-coded	*sipA*, *sopE*	[25]
	*S.* Typhimurium	InvS	Trans-coded	PrgH, FimZ	[26]
	*Salmonella*	PinT	Trans-coded	*hilA*, *rtsA*	[27]
Resisting Oxidative Stress	*S.* Enteritidis 2472	Sal-1	Trans-coded	iNOS	[28]
	*S.* Typhimurium	OxyS	cis-coded	*ompX*	[29]
	*S.* Typhimurium	CyaR	Trans-coded	*ompX*	[29]
	*S.* Typhimurium	MicA	Trans-coded	*ompX*	[29]
	*S.* Typhimurium *YK5104*	RaoN	Trans-coded	*ldhA*	[29]
	*S.* Typhimurium	RyhB-1, RyhB-2	Trans-coded	-	[23,30]
Survivalin Macrophages	*S.* Typhimurium	IsrC	cis-coded	*msgA*	[23]
	*S.* Typhimurium	IsrN	cis-coded	*STM2765*	[23]
	*S.* Typhimurium	OxyS	cis-coded	*-*	[23]
	*S. Typhi*	RfrA, RfrB	Trans-coded	*-*	[31]
	*S.* Typhimurium	RyhB-1, RyhB-2	Trans-coded	*fumA*, *sdhD*	[32]
	*S.* Typhimurium	IsrM	Trans-coded	SopA	[24]
	*S.* Typhimurium *YK5104*	RaoN	Trans-coded	*-*	[30,33]
Regulating Inflammatory Cytokines of Hosts	*S.* Typhimurium	PinT	Trans-coded	IL-8, SOCS3	[34,35]

## Data Availability

Not applicable.

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
