# Peer review of "Functions of Small Non-Coding RNAs in Salmonella–Host Interactions"

_biology, 2022, doi:10.3390/biology11091283_

Round 1

Reviewer 1 Report

The review article entitled “Functions of small non-coding RNAs in Salmonella host interactions” by Xia Meng and colleagues is very well written and compiled very good literature. This review article nicely describes the role of Salmonella encoded sRNAs in host-pathogen interactions.  This review provides the necessary information on how these sRNAs work for betterment of Salmonella survival in host. However, there are some minor points that authors may want to describe in this review if possible:

1.     How DsrA modulates stress response? Does deletion of DsrA affects the expression of RpoS?

2.     How NADH dehydrogenases helps in maintaining acid tolerance in Salmonella by 6S RNA. Does it provide some metabolic shift in Salmonella?

3.     STnc640 upregulates the expression of fimA, then how it decreases the adhesion ability of S. enteritidis? It contradicts the function of STnc640. Is there any mechanism or proposed mechanism behind this type of phenomenon?

4.     Intestinal hyperpermeability of host is often reported with salmonella infection. Is there any sRNA from Salmonella or E. coli reported that could directly or indirectly alter the gut permeability of the host?

Reviewer 2 Report

This review systematically introduces the important role of non-coding RNAs in adapting to the host environment. The writing of this manuscript is very standard and rigorous. It is recommended for acceptance. However, the legend in Figure 1 could add some description to this conclusion figure.

I have only two small suggestions: 1. Figure 1 is a little fuzzy under magnification, so it is suggested to improve the resolution of the picture; 2. The legend in Figure 1 contains only one sentence. It is suggested that a description of this figure should be added to the legend as a summary of the whole paper. Thanks.
